# Deciphering Post-Stroke Sleep Disorders: Unveiling Neurological Mechanisms in the Realm of Brain Science

**DOI:** 10.3390/brainsci14040307

**Published:** 2024-03-25

**Authors:** Pinqiu Chen, Wenyan Wang, Weikang Ban, Kecan Zhang, Yanan Dai, Zhihong Yang, Yuyang You

**Affiliations:** 1Key Laboratory of Molecular Pharmacology and Drug Evaluation, Ministry of Education, Collaborative Innovation Center of Advanced Drug Delivery System and Biotech Drugs in Universities of Shandong, School of Pharmacy, Yantai University, Yantai 264005, China; 15695197298@163.com (P.C.);; 2Institute of Medicinal Plant Development, Chinese Academy of Medical Sciences and Peking Union Medical College, Beijing 100193, China; 3School of Automation, Beijing Institute of Technology, Beijing 100081, China

**Keywords:** sleep disorders, stroke, mechanisms, brain regions, circadian rhythms, sleep stage, neurotransmitter, inflammation, ion, kinase, treatment strategy

## Abstract

Sleep disorders are the most widespread mental disorders after stroke and hurt survivors’ functional prognosis, response to restoration, and quality of life. This review will address an overview of the progress of research on the biological mechanisms associated with stroke-complicating sleep disorders. Extensive research has investigated the negative impact of stroke on sleep. However, a bidirectional association between sleep disorders and stroke exists; while stroke elevates the risk of sleep disorders, these disorders also independently contribute as a risk factor for stroke. This review aims to elucidate the mechanisms of stroke-induced sleep disorders. Possible influences were examined, including functional changes in brain regions, cerebrovascular hemodynamics, neurological deficits, sleep ion regulation, neurotransmitters, and inflammation. The results provide valuable insights into the mechanisms of stroke complicating sleep disorders.

## 1. Introduction

Stroke is the leading cause of death and disability globally and seriously affects the health of adults. Studies highlight that those who experience stroke often face an increased risk of a subsequent occurrence, with worldwide incidence rates increasing. The disease burden of stroke has dramatically increased over the previous 30 years, with the incidence of stroke increasing by approximately 70.0% and the number of stroke deaths increasing by approximately 43.0% [1]. The highest incidence of stroke is found in low- and middle-income countries, especially Eastern Europe, Asia and sub-Saharan Africa, notably the United Arab Emirates (208.2 per 100,000) and China (144.8 per 100,000). Even in economically better countries, the incidence is high as 40.4 per 100,000 (New Zealand) [2]. A notable detail is that prevalent post-stroke sleep disturbances, with rates ranging from 76% to 82%, exhibit distinct clinical features [3,4]. Current research revealed that as many as 92.4% of patients with mild to moderate ischemic strokes suffer from acute sleep disorders, underscoring the concerning sleep health of cerebral infarction patients [5]. Studies have indicated that sleep disorders, such as insomnia, narcolepsy, heteromorphic sleep, circadian rhythm disturbances and periodic limb movement disorders have high potential to contribute to the prognosis of stroke [6]. After a stroke, the prevalence of sleep apnea is notably high, with estimates reaching as much as 70% [7]. Factors amplifying this trend include aging demographic and stroke-associated risk elements, which increase the cumulative disease burden [8,9]. This situation requires high-level attention, and robust measures should be taken to prevent stroke and its complications to improve the quality of life of patients. Although emerging research has discerned a correlation between sleep anomalies and the evolution of ischemic stroke, suggesting a bidirectional interaction, the molecular intricacies bridging them remain partially veiled [10,11]. The academic realm is progressively accentuating the diagnosis and therapeutic approaches for sleep irregularities post-stroke, a step pivotal for the comprehensive recuperation of patients and their families. Therefore, an in-depth probe into the genesis of post-stroke sleep disturbances is of monumental value. This study aimed to demystify the reciprocal relationship and inherent biological mechanisms tethering post-stroke sleep anomalies to cerebral ischemia. The areas of emphasis included neurotransmitter modulation, sleep patterns, neurovascular entities and inflammatory reactions, aiming for a holistic understanding of sleep anomalies post-stroke (Figure 1). This study concurrently focused on the latest therapeutic advances in post-ischemic sleep disorders to furnish innovative insights and therapeutic trajectories for clinical application. Whilst advancing measures for the prevention and mitigation of ischemic stroke progression remains crucial, sleep indisputably serves as an effective prospective intervention target aimed at enhancing the prognosis of patients with ischemic stroke [12,13].

### Method

The aim of this literature review is to explore the effects of SD on memory function and to advance the field by describing relevant mechanisms. We conducted electronic searches using reputable databases, including PubMed, Google Scholar, Web of Science, and CNKI, to identify recent publications. The search terms employed were: Sleep Disorders AND Stroke; Function AND Mechanisms; Insomnia AND Stroke; Sleep Deprivation AND Stroke Impairment; Circadian Rhythms AND Mechanisms of Injury; Stroke AND Sleep Stage; Stroke AND Inflammation; Stroke AND Treatment Strategy. Emphasis was placed on recent publications, and subsequently, we scrutinized the reference sections of pertinent articles to uncover older publications aligning with our research goals. Ultimately, we selected the most recent articles that offered contemporary and pertinent information on our predefined subtopics.

The present review is structured into the following sections: (a) Biological Mechanisms Underlying Sleep Disorders Following Stroke; (b) Recent Treatment Progress (Synaptic Plasticity, Neurons, Oxidative Stress, Genes, Neurotransmitters, Circadian rhythms, Rodent to human complexities); (c) Recent Treatment Progress; and (d) Conclusion and Prospect.

## 2. Biological Mechanisms Underlying Sleep Disorders following Stroke

### 2.1. Changes in Sleep Staging

Changes in brain activity and sleep activity during stroke are associated with poor patient outcomes, and correct assessment of such activity changes is critical [14,15,16,17,18,19]. The disruption of sleep staging is a feature of post-stroke sleep. Previous studies indicated pronounced shifts in sleep patterns amongst patients with cerebral infarction, characterized by prolonged non-rapid eye movement (NREM) durations, suppressed rapid eye movement (REM) sleep and varying sleep patterns contingent upon the hemisphere affected by the lesion [19,20]. Human and animal subjects exhibited a marked reduction in REM sleep following acute ischemic strokes, coupled with notably diminished sleep efficiency (SE) and altered sleep architecture. In comparison to a control group, those with acute cerebral infarction displayed significant reductions in total sleep time (TST), SE and REM phases (*p* < 0.01). Conversely, the wake-after-sleep onset duration exhibited a notable surge (*p* < 0.01) [21,22]. Studies indicated that individuals who have experienced strokes typically exhibit reduced TST compared with their healthy counterparts. This reduction in TST is more pronounced in individuals with a history of recurrent strokes [3]. A study juxtaposing polysomnography outcomes between 51 patients with stroke and 21 unafflicted controls discerned a substantial TST decrease in a stroke cohort [23]. Pace et al. [24] observed that in the acute phase of stroke, patients with poor short-term functional outcomes exhibited reduced REM sleep and prolonged REM onset latency. In certain instances of supratentorial stroke, a transient decrease in REM sleep may be observed, with a higher frequency noted in cases of right-sided strokes [25]. Animal studies have demonstrated that ischemic stroke can selectively suppress REM sleep in mice, leading to a notable reduction in REM sleep [26].

### 2.2. Circadian Disruption

Evidence suggests that stroke can disrupt endogenous circadian rhythms by affecting the suprachiasmatic nucleus or neuron-associated clock mechanisms [27,28]. In acute stroke, the physiological circadian rhythm of blood pressure changes, displaying a loss of biphasic circadian rhythm variability compared with individuals with normal blood pressure [29]. Naturally, blood pressure typically decreases by around 10% or more during the night [30]. A study involving 50 patients with acute stroke identified a loss of blood pressure circadian rhythm in these individuals [31]. Post-stroke, circadian rhythms are altered, especially in the sleep/wake cycle, leading to sleep fragmentation and decreased SE [27,32]. Such alterations closely correlate with the development of post-stroke apathy [33]. Additionally, ischemic injury induces an advancement in the phase of Per1 expression and disrupts the rhythm of melatonin secretion, thereby modulating Bmal1 expression. Both of these processes are crucial for cell survival in neuronal ischemia [34,35].

Blood angiogenic regulators play a key role in modulating endothelial and pericyte function, and they are crucial to the angiogenic process in ischemic stroke. Vascular endothelial growth factor (VEGF) is a downstream target of the circadian clock network, and VEGF proteins revealed an oscillatory expression pattern regulated by the core circadian components *Bmal1*, Clock, Per and Cry genes [36]. Overexpression of *Bmal1* promoted the luciferase activity of VEGF and the knockdown of VEGF expression. However, it reversed the promotional role of Bmal1 in promoting HUVEC angiogenesis [37].

Taken together, these results showed that the onset of stroke leads to the desynchronization of endogenous biological rhythms and disrupts the expression pattern of biological clock genes, suggesting that interventions in the circadian system, such as environmental modifications, chronotherapy and targeting clock genes, may be a potential target for post-infarction therapy.

### 2.3. Functional Changes in Brain Regions

Evidence indicates that stroke injures sleep-regulating brain regions, subsequently inducing sleep disturbances (Table 1). Sleep disorders may be attributed not only to the neurobiological changes induced by cerebral ischemia but also to the disruption of sleep regulatory pathways. Experts categorized brain structures linked with sleep into wakefulness-promoting, sleep-inducing and REM systems, suggesting that damage to these areas might precipitate sleep complications. Investigations pinpointed that stroke-induced lesions in the frontal lobe, basal ganglia and brainstem substantial increased the likelihood of post-stroke sleep disorders, underscoring the potential link between specific brain region damage and ensuing sleep disruptions [8,38]. In a study comprising 508 patients with ischemic stroke, where stroke lesions were classified into distinct brain regions, the involvement of the frontal lobe emerged as a predictive factor for sleep disorders [39,40]. The research further indicated the susceptibility of the frontal lobe to ischemic changes in the subcortical region. A study delineated variations in sleep disorder incidences based on infarct locations. Brainstem infarctions topped the list with 70.40% incidence, followed by basal ganglia at 69.0%, cerebral cortex at 41.30%, cerebellar at 36.80% and thalamic infarctions at 16.70% [41]. Another investigation corroborated this trend, pinpointing the brainstem as the most susceptible region, followed by the thalamus, cerebral hemisphere, basal ganglia and cerebellum [42,43]. The implicated areas play pivotal roles in governing sleep and wakefulness cycles. Further exploration targeted stroke locations and unveiled that patients with cerebral cortex (36.67%) and subcortical cortex (53.33%) lesions were more prone to sleep disturbances than cerebellum-afflicted patients (10.0%) [44]. This research also highlighted potential ties between post-stroke damage in specific brain regions, such as the anterior occipital horn, lateral thalamus, posterior insula and medial temporal lobe, and circadian rhythm sleep disorders [45]. Hcrt, which is predominantly synthesized by lateral hypothalamic neurons, extensively projects across the CNS, modulating sleep and influencing excitability in diversified brain areas [46]. The preoptic (PO) hypothalamus is vital for NREM and REM sleep genesis and NREM homeostasis, with both sleep phases partially orchestrated by this region [47]. Another study posited that post-acute cerebral infarction sleep disturbances may be correlated with thalamic infarct locations. This potential association is suggested to arise from the interference of IL-17 and Hcrt overexpression with thalamic operations [3,48]. Additionally, patients with thalamic stroke manifested diminished nocturnal slow wave sleep (SWS), hinting at synaptic impairments within the thalamus. Overall, stroke causes a significant impact on the microstructure of sleep.

### 2.4. Cerebral Vascular Hemodynamics

Cerebral ischemia alters cerebrovascular hemodynamics, thus affecting oxygen and nutrient supply to the brain, with profound implications for nervous system function [62]. This impairment transcends cognitive and motor functions, extending into the modulation and stage of sleep [63,64]. Recent studies highlighted the intricate link amongst cerebral ischemia, cerebrovascular hemodynamics and sleep [65,66].

Upon cerebral ischemia onset, local brain tissue blood flow sharply reduces or completely ceases [67]. Reperfusion restores oxygen, counteracting cerebral hypoxia effects, yet ischemia/reperfusion may potentially amplify disease progression [68]. Numerous studies have documented the relationship between cerebral blood flow and sleep alterations in humans and rodents [69,70,71,72,73]. During SWS, cerebral blood flow slightly diminished compared with that in wakeful state [73]. Using two-photon microscopy, a study observed a significant surge in capillary cerebral blood flow (CBF) during REM sleep across various cortical regions. REM sleep deprivation led to increased capillary red blood cell flow, emphasizing the crucial role of CBF in REM sleep regulation, notably mediated by adenosine A2a receptors [74]. In terms of cerebral blood flow velocity, an initial increase followed by a decline was observed during sleep [75]. Clinical analyses have showcased the link between suboptimal sleep quality and a decline in cerebral blood flow in areas like the right orbitofrontal and insular cortex [76]. Individuals with NREM disorders manifested reduced CBF in regions such as the parietooccipital lobe (anterior cuneiform lobe), marginal gyrus and cerebellar hemisphere [77].

In a three-dimensional realm, the mean distance from a neuron cell’s center to its closest microvessel is 15 μm [78]. The extracellular space is fluid, adapting to heightened brain activity and sleep [79]. Surrounding the cerebral blood vessels are various cell types, emitting substances into the extracellular domain and adjacent to blood vessels, orchestrating cerebral blood flow. These cell types, encompassing endothelial cells, pericytes, astrocytes, perivascular macrophages and peripheral microglia, are collectively termed neurovascular units (NVUs) [80,81] (Figure 2). NVU meticulously orchestrates the regulation of the blood–brain barrier, CBF, blood flow velocity and regional cerebral blood volume, ensuring the stability of the brain’s microenvironment [82,83,84,85,86].

NVUs serve a pivotal role in both the pathophysiological progression of ischemic stroke and the modulation of cerebral blood flow during states of sleep and wakefulness. After cerebral ischemia, NVUs and their microvessels undergo diverse extents of injury [87]. This phenomenon encompasses structural alterations and related cellular dysfunctions, which subsequently influence the regulation of cerebral blood flow [88]. Consequently, these affect neural functioning and sleep patterns.

The dysfunction of NVUs can be directly instigated in the initial stages of ischemia. Amongst the components of NVU, endothelial cells are the foremost to sustain damage in the ischemic brain region. The dysfunction is closely trailed by the activation of microglial cells and astrocytes. Once activated, these cells begin secreting pro-inflammatory mediators, such as tumor necrosis factor (TNF) and interleukin (IL)-1β, and witness an upsurge in the expression of IL-6 [89,90]. A notable detail that pro-inflammatory cytokines, including but not limited to IL-1β and TNF-α, possess the capability to modify vascular hemodynamics [91,92,93].

### 2.5. Neurological Deficits

Numerous studies highlighted that the onset of sleep disorders following a stroke is intricately linked to the severity of a patient’s neurological deficits [94]. One study indicated that patients with thalamic strokes encountered a decrease in their slow-wave sleep during nighttime, pointing towards damage to the synapses of the thalamic nerves [95]. The findings of the study emphasized the differences in sleep disorder incidences amongst patients with varying degrees of neurological deficiencies. Specifically, 46.1% of those with mild neurological deficits faced sleep disorders, and the number surged to 73.7% for those with moderate to severe deficits [41]. National and international research echoed that patients with post-stroke sleep disorder tend to have intensified neurological deficits. Research denoted that those with more severe neurological issues witnessed a decrease in REM sleep duration [96]. Consequently, the severity of these deficits positively correlates with the incidence of sleep disturbances, implying that profound neurological impairments result in an increased likelihood of sleep disorders [96].

### 2.6. Sleep Disruptions: Exploring Ions, Ion Channels and Kinases

#### 2.6.1. Ions

The underpinnings of sleep disorders following cerebral ischemia are profoundly associated with ion interference regulation [97]. Specifically, cerebral ischemia may disrupt ion channel functions and ion equilibriums, thus perturbing sleep regulation processes [98,99,100]. Sleep is intrinsically regulated by ion channels, with key ions like potassium (K^+^), calcium (Ca^2+^) and magnesium (Mg^2+^) playing crucial roles [99,101,102]. Reports indicated that the activation of the G protein-gated inward rectifier K^+^ channel (GIRK) can foster NREM sleep and solidify it [102]. By examining cerebrospinal fluid in healthy individuals during sleep, wakefulness and sleep deprivation, a study found a significant decrease in K^+^ concentrations in sleep and sleep-deprived states compared with wakefulness [97]. Concurrently, GIRK governs the circadian excitatory rhythm of brain neurons, with the regulatory process hinging on alterations in extracellular K^+^ concentration [103]. Studies have shown that impairments to the Kcnk9 channel can diminish sleep durations [104], and the two-pore K^+^ channel (K2P) is pivotal in regulating the sleep-wake cycle [105]. Intriguingly, ORK1 (a human TREK1 analogue) was identified to modulate the sleep duration in fruit flies [106]. A notable detail that the arousal mechanism is independent of the AMPA receptor but interlinked with a surge in extracellular K^+^ concentrations [107]. NMDA receptor deficiencies in LPO neurons curtailed NREM and REM sleep and resulted in substantial sleep-wake fragmentation [108]. During REM sleep, the LPO region sees a selective augmentation in calcium activity via NMDA receptors [109]. Genetically silencing NMDA receptors in the brain truncates the total sleep duration, a phenomenon evident in fruit flies and rodents [110,111]. Studies have elucidated that awake states led to a decrease in extracellular Ca^2+^ and Mg^2+^ levels [107]. ASICs, which are proton-activated non-voltage-dependent ion channels, play a role in regulating sleep-wake cycles [112]. The nucleus of the blue patch (LC) is a key brain center in sleep-wake regulation. Recent studies underscored the significance of ASICs in LC neurons, particularly during the shift from NREM to REM sleep states [113]. Certainly, ion channels, associated ions and ion receptors play pivotal roles in the regulation of sleep.

#### 2.6.2. Ion Channels

Cerebral ischemia instigates intricate reactions within the neurovascular unit system, resulting in detrimental effects on vascular endothelial and neural cell functions. This phenomenon sets off a cascade of pathological alterations, with ion channels being at the epicenter of these changes. These neuronal ion channels have been identified as significant players in ischemic cerebrovascular conditions [114]. When cerebral ischemia occurs, the irregular opening and closing of ion channels due to ischemia and subsequent reperfusion, and the disequilibrium of ion balance within and outside neurons emerge as vital factors in ischemic brain injury [115]. Amongst these channels, the TREK-1 channel, predominantly expressed in the central nervous system, is closely intertwined with cerebral ischemia. This channel becomes activated during cerebral ischemia due to several factors: neuronal hypoxia, glucose deprivation, cell swelling and the liberation of excitatory amino acids [116]. Under normal circumstances, the concentration of intracellular calcium ions remains relatively consistent. However, during the injury phase of cerebral ischemia–reperfusion, hypoxia leads to the aberrant activation of extracellular calcium ion channels. This aberration prompts a swift increase in intracellular Ca^2+^ concentration. Such an increase instigates phospholipase C to produce IP3, which subsequently activates the IP3R calcium channel. This series of actions culminates in the sarcoplasmic reticulum discharging copious amounts of calcium ions into the cytoplasm, leading to calcium overload [117]. Furthermore, an excessive secretion of the excitatory neurotransmitter glutamate occurs in the aftermath of ischemia, precipitating cellular death. This phenomenon is intrinsically connected to a surge in the intracellular Ca^2+^ concentration [118]. Research has illuminated that the calcium overload observed post-cerebral ischemia–reperfusion is attributable to an extended opening duration, heightened opening probability and an amplified channel current magnitude of L-type calcium channels [117].

ASIC comprises seven subunits, with ASIC1a homodimer channels and ASIC1a/2b channels playing significant roles in cerebral ischemic injury [119]. Cerebral ischemia triggers the generation of H^+^ through processes like lactate accumulation and ATP hydrolysis, resulting in tissue acidification. Extracellular acidification activates ASICs, notably ASIC1a channels, leading to increased Ca^2+^ influx and subsequent neuronal cell death [120]. Research indicated that the downregulation of ASIC2a is associated with compromised neurological function and increased infarction rate in rats experiencing cerebral ischemia. The mechanism involves ASIC2a reducing the inhibition of calcium ion entry into neurons, thereby facilitating an increased influx of calcium ions into the cells [121].

#### 2.6.3. Kinases

Genetic studies highlighted the importance of several kinases, such as ERK1/2, CaMKII/β and SIKs, in promoting sleep. Drosophila studies showed that ERK functions as a promoter of sleep, and sleep deprivation led to an increase in ERK phosphorylation [122]. A rat study demonstrated that sleep deprivation resulted in a decrease in ERK1/2 expression [123]. Subsequent trials involving mouse embryos and conditional knockout experiments provided evidence supporting the involvement of ERK1/2 in the regulation of the natural sleep-wake cycle [124]. Research revealed that mice with knockout mutations in Camk2a or Camk2b displayed a markedly reduced sleep duration and a decreased likelihood of transitioning from sleep to wakefulness [125,126]. Additionally, CaMKII displays a brain region-specific role in sleep regulation. The SIK family contains three isozymes, SIK1, SIK2 and SIK3. Often referred to as the “sleep gene”, SIK3 is notably the most abundant in the brain [127]. SIK3 knockout mice exhibited periodic dysregulation of circadian rhythms [128,129]. Disrupting circadian rhythms and PER protein cycling were observed in Drosophila neurons when SIK3 was knocked down, underscoring the influence of SIK3 on these biological patterns [130]. In addition, the sleep-promoting role of SIK3 was conserved in Drosophila and *Hidradenitis elegans* nematodes, suggesting that sleep regulation is conserved in invertebrates [131]. Interestingly, contrasting results emerged, indicating that mice lacking SIK3 or LKB1 experienced reduced sleep. This reduction was primarily attributed to heightened SIK3 signaling in excitatory neurons in the cerebral cortex and hypothalamus. This finding manifested as an increase in EEG delta-wave power during NREM sleep, along with an extended duration of NREM sleep [132,133,134]. The findings strongly suggested that kinases, such as ERK1/2, CaMKII/β and SIKs, actively contribute to the homeostatic regulation of sleep quantity and the underlying demand for sleep.

CaMKII displays heightened sensitivity to ischemia. In the wake of cerebral ischemia, an overload of intracellular Ca^2+^ triggers the activation and augmented autophosphorylation of CaMKII [135]. This cascade results in a significant increase in p-CaMKII and a pronounced reduction in non-phosphorylated CaMKII, culminating in diminished CaMKII activity [136,137,138]. In vivo, the functions of diverse Ca^2+^-dependent kinases are governed by serum calmodulin (CAM). After cerebral ischemia, the levels and activity of CAM increase concomitantly with elevated intracellular Ca^2+^ concentrations in neuronal cells. Importantly, this upregulation of CAM correlates positively with the severity of the ischemic condition [139]. Clinical investigations have revealed that CAM levels in patients with ischemic stroke reached (189.34 ± 24.98) ng/mL, a significant elevation compared with the control group’s level of (80.92 ± 20.59) ng/mL [140]. Intense intracellular Ca^2+^ influx can lead to the formation of Ca^2+^/CaM complexes [141]. Under hypoxia–ischemia, the overactivated Ca^2+^/CaM complex, which binds and regulates multiple downstream target enzymes, causes excessive apoptosis of brain cells, and the mechanism is related to the over-activation of the CaM/CaMK II signalling pathway [142,143].

The mechanism of decreased SIK2 expression and increased neuronal cell injury after cerebral ischemia/reperfusion in rats is related to decreased HIF-1α expression, decreased ATP content and increased ADP content in brain tissue compared with ischemia [144]. A study found that SIK2 expression in the brain significantly decreased after hemorrhagic stroke [145].

ERK1/2 is a family member of mitogen-activated protein kinases [146]. Several studies have indicated P-ERK1/2 changes after ischemia in ischemic models. Previous studies have shown that in a rat MCAO model, P-ERK1/2 increased significantly in the ischemic semi-dark zone area and infarct focal area at 90 min postischemia and reperfusion for 1–3 h. Amongst them, P-ERK1/2 peaked at 1 h after 90 min of reperfusion after ischemia, and its expression was significantly higher in the ischemic semi-dark zone than in the infarct focal area [147]. A study on a mouse animal model of MCAO found that P-ERK1/2 expression decreased in the infarct region after ischemic administration of the MEK1-specific inhibitor PD98059 [148]. Further studies revealed that inhibition of the ERK1/2 pathway significantly inhibited Cx40/Cx43 heterodimeric junctions and NF-κB and improved neurological function in rats [149].

In brief, stroke may disrupt sleep by directly or indirectly affecting sleep-related ion concentrations, ion channels and kinase expression.

### 2.7. Neurotransmitter Regulation

The etiology behind sleep disturbances in patients with stroke is multifaceted. Sleep disorders post-stroke have intricate interrelations with neurotransmitters. Brain cell damage from the stroke can interfere with the secretion of sleep-associated neurotransmitters or other relevant factors. Neurotransmitters, which can be influenced by physiological disruptions such as cerebral ischemia, can in turn affect sleep patterns. A notable detail is that sleep disturbances in patients with stroke may stem from the anomalous secretion of neurotransmitters, particularly a reduction in serotonin (5-HT). Some experts postulated that post-stroke sleep disorders could be linked to causing harm to serotonin neurons in the affected area, inhibiting the serotonin neurotransmitters’ normal function. Key neurotransmitters linked to sleep, such as 5-HT and norepinephrine, may undergo secretion anomalies when stroke occurs, mainly showing a decline in secretion, potentially culminating in sleep disruptions. Studies have revealed that serum concentrations of 5-HT and NE in stroke patients with sleep disorders are below the standard and linked with the severity of the ailment [150]. Furthermore, research has suggested that low-frequency electrical acupoint stimulation may be a viable treatment for post-stroke sleep disorders, potentially by boosting plasma 5-HT levels and decreasing plasma NE levels [151]. The efficiency of 5-HT receptor blockers in reducing mortality and bettering the long-term outlook for post-stroke patients with depression and anxiety underscored the integral connection between post-stroke depression and 5-HT [42]. Thus, post-stroke sleep disturbances may be intertwined with a drop in 5-HT levels. If the stroke-damaged region affects 5-HT neurons, causing reduced secretion of 5-HT, it may precipitate sleep issues.

Besides 5-HT, melatonin is a neurotransmitter intrinsically linked to sleep. Studies underscored melatonin’s favorable influence on enhancing sleep quality. Notably, patients with stroke demonstrated an irregular melatonin secretion rhythm at nighttime, marked by a nocturnal elevation and diurnal decline. Specific figures indicate that in contrast to the melatonin levels of healthy individuals (86.14 ± 24.94 pg/L, those of patients with stroke drastically decreased to 54.08 ± 33.33 pg/L 2 days post-onset, gravely impairing their sleep [152]. Another key player in sleep regulation is the small molecule neuropeptide orexin.

Recent clinical research has delved into the changes and effects of serum orexin (OxA) concentrations in cerebral infarction patients post-onset, but findings have been varied. However, the diminished cerebral perfusion arising from acute cerebral ischemia in a patient’s stroke is acknowledged to curtail their capacity to cope with emergent scenarios, subsequently augmenting orexin expression levels and potentially propelling the patient into an extended wakeful state. Such insights are pivotal in fostering a more profound comprehension of the biological underpinnings of post-stroke sleep disturbances. Recent research confirmed that OxA levels increased during cerebral ischemia, providing notable neuroprotective effects [153,154]. Cerebral ischemia prompts the defensive activation of OxA, escalating its concentration. This increase augments OxA’s inherent expression levels, causing patients to remain alert [155]. Clinical research indicated that patients with ACI have markedly increased serum OxA levels compared with controls [156]. Another investigation into patients with sleep disorders had significantly higher OxA levels than normal (63.42 ± 37.56 vs. 54.84 ± 23.95 pg/mL) [157]. Studies suggested that transcranial repetitive needle stimulation may alleviate post-stroke sleep disorders by reducing serum OxA levels [157,158].

Past findings revealed that in rats with a middle cerebral artery occlusion reperfusion injury, the OxA expression in the brain tissue was markedly greater on the ischemic side than on the non-ischemic counterpart [159]. Notably, an optimal OxA dosage can appreciably diminish cerebral infarction zones, underscoring its neuronal protective capacity [160]. This trend has been confirmed in other studies, which showed that OxA significantly improved neurological deficit scores and infarct volume after cerebral ischemia–reperfusion, with 30 μg/kg showing the best results [161]. Moreover, a controlled study on ataxin-3 transgenic mice (with OxA gene knockout) in a middle cerebral artery occlusion model revealed that the knockout mice exhibited greater brain damage volume at 24 and 48 h post-occlusion and deteriorated neurological function scores than the wild-type mice [162]. Chronic cerebral ischemia has also been pinpointed as the reason for the increase in orexin mRNA expression [163].

Within the rat cerebral cortex, orexin substantially boosts neuronal survival in a dose-dependent manner, with this pro-survival trait linked to reduced caspase-3 activity. Researchers identified that OxA treatments notably modified the mRNA expression levels of TNF-α and IL-6. OxA effectively ameliorated the comatose state in brain-injured rats, with the efficacy being concentration-driven. This mechanism may be tied to the increased RasGRF1 protein expression in the prefrontal cortex. In essence, OxA concentrations increase during cerebral ischemia, generally safeguarding neurons. Some studies indicated that sleep disorders in rats with chronic cerebral ischemia stem from heightened OxA neuron activity in the brain, which heightened wakefulness and induced sleep disturbances [164,165]. Furthermore, adenosine, a potent innate sleep-enhancing agent, diminishes in the brain during sleep deprivation and surges during sleep recuperation.

### 2.8. Inflammatory Cytokines

Inflammation resulting from cerebral ischemia may be a potential trigger for associated sleep disorders. Studies have shown that individuals with inflammatory diseases are often more prone to sleep disturbances [166,167]. In the context of stroke, inflammation disrupts the balance between pro-inflammatory and anti-inflammatory reactions. Patients with sleep disorders due to acute cerebral infarction displayed increased inflammatory factors, including C-reactive protein (CRP), IL-6, IL-1β, TNF-α and Hcrt, compared with normal levels [48,168]. Earlier animal research found that in post-transient cerebral ischemia, gerbils showed increased levels of, IL-6, IL-1β and TNF-α, especially in the initial stages [169]. REM sleep is suppressed during times with elevated inflammatory mediators like cytokines.

A strong link exists between CRP and IL-6 levels and intravascular inflammation [170]. Elevated levels of pro-inflammatory markers CRP and IL-6 are tied to sleep disturbances [170,171]. Typically, pro-inflammatory cytokines, including IL-1β, TNF-α, IL-18 and IL-6, promote NREMS [172,173]. Studies indicated that IL-1β can boost NREM sleep and diminish REM sleep in animals. This effect may be attributed to the modulation of sleep homeostasis through changes in inflammatory cytokine and circadian gene expression [174]. Sleep deprivation-induced IL-6 expression alterations can affect the sleep cycle by affecting NREM [175,176]. Moreover, injecting IL-6 into healthy subjects prolonged the NREM phase, particularly NREM phase 3, accompanied by an increase in CRP levels [177]. Persistent sleep deprivation or brief periods led to a notable increase in CRP levels [178]. Hence, IL-6-induced sleep disturbances may be related to the activation of the hypothalamic IL-6 signaling pathway, promoting the release of hormones like corticosteroids and cortisol [179]. Increased serum levels of CRP and IL-6 correlate with stroke severity; notably, patients with moderate to severe strokes generally present higher CRP and IL-6 levels [171,180]. These findings suggested that post-cerebral ischemia inflammatory factors particularly influence the NREM sleep cycle.

The IL-1 cytokine family, encompassing IL-1α, IL-1β, IL-18, IL-33, IL-36α, IL-36β, IL-36γ, IL-1RA, IL-36RA and others, plays roles in inflammation and sleep regulation [181,182]. Disorders, such as brain injury and cardiovascular disease, have links with IL-1 family members [169,183]. Research indicated the IL-1 family affects sleep by modulating downstream receptor activity. Sleep deprivation, ongoing sleep restriction and infections increase IL-1β levels in various brain areas, associated with NREM sleep and SWA. Sleep-disturbed animals, including rabbits, cats, monkeys, mice and rats, along with several human brain areas, peripheral tissues and circulating IL-1β, showed increased expression levels [173]. In cerebral ischemia animal models, the IL-1β mRNA expression spiked within minutes of ischemia onset [184], and its protein expression surged 4–6 h later. Interestingly, IL-1α expression preceded IL-1β expression [185], emphasizing the IL-1 family’s influence on sleep. Studies identified IL-1β’s role in impeding neurogenesis, closely linked to inflammation-induced neurological deficits, mediated by endogenous ligands through TLRs [186]. TLR4-mediated neuroinflammation plays a key role in the pathogenesis of secondary sleep disorders after stroke [186,187]. TLR4-mediated neuroinflammation is pivotal in secondary sleep disorder pathogenesis following a stroke. Activating TLR4 initiates the NF-κB pathway and NLRP3 inflammasome, causing a subsequent increase in the pro-inflammatory cytokine IL-1β, closely tied to post-stroke sleep disturbances [188,189,190]. NLRP3 activation via the NF-κB and TLRs’ pattern recognition receptors and the formation of NLRP3 inflammasomes can induce IL-1β production via Caspase-1 [191,192,193]. The NF-κB/NLRP3 pathway, indicated by changes in protein levels (TLR4, NLRP3, IL-1β, and Caspase-1), is linked to neurocytogenesis and angiogenesis [191]. This implies its role in post-stroke sleep disorders.

TNF-α is the most extensively studied member of the TNF family; it has complete sleep regulation functions, and its expression changes are closely related to sleep disorders [194,195]. TNF-α can be activated by NF-κB to trigger various inflammatory signal transduction pathways. NF-κB as a transcription factor B can activate NOS, COX-2 and adenosine A1 receptors, act on brain regions related to sleep regulation (hypothalamic preoptic area and basal forebrain), and affect brain regions related to sleep regulation, such as the hypothalamic preoptic area and basal forebrain [196]. Meanwhile, TNF-α promotes NREM sleep and interferes with the sleep awakening rhythm by regulating the growth hormone system and HPA axis [197,198,199]. This effect is connected to circadian rhythms and sleep periods. TNF-α can alter CLOCK-BMAL1 activation, indicating that the circadian rhythm can be altered through TNF-α adjustments [200]. In the acute phase of cerebral ischemia, the proliferation and activation of microglia led to a robust inflammatory response [89], characterized by increased levels of TNF and IL-1β and upregulation of IL-6, which can result in sleep disorders.

The disorder of NVUs may be one of the important mechanisms of inflammation and sleep disorders caused by cerebral ischemia. NVU directly affects neuroinflammatory responses and sleep conditions by tightly and rapidly regulating the cerebral microvascular system. The damage to NVUs can disrupt their homeostasis, leading to disrupted release of inflammatory molecules and ultimately leading to sleep disorders.

### 2.9. Challenges and Opportunities

Research on post-stroke sleep disturbances, both domestically and internationally, often employs overlay compound models to replicate ischemia-induced sleep disorders (e.g., MCAO+PCPA, MCAO+MMPM) [165,201]. While these models closely mimic symptoms post-stroke, there are notable disparities from actual sleep disturbances in human stroke survivors. It is crucial to acknowledge that commonly used rodent models, such as mice, though extensively studied, still have limitations in understanding their brain development. Compared to humans, rodent brains are significantly smaller, structurally simpler, and exhibit lower cognitive abilities [202]. Hence, emphasizing human studies and cross-species comparisons is essential to overcome challenges in translating research findings from animal models to humans.

Despite being considered the gold standard for assessing sleep [203], current EEG monitoring tools still have certain limitations. The abundance of monitoring systems and inconsistent sleep staging standards necessitates a systematic review and analysis of relevant research [204]. While commercial sleep monitoring devices exist, their accuracy requires improvement [205]. In comparison to polysomnography, studies have not yet reached clinical standards, highlighting the need to balance performance, functionality, cost, and user-friendliness [206]. Biomechanical signal monitoring techniques typically relate to specific behaviors such as teeth grinding, restless legs, and breathing [207].

On the other hand, understanding the correspondence between EEG data and human behavior faces challenges due to conceptual ambiguity. Additionally, EEG data collection is susceptible to external environmental interference. To address these issues, it is recommended that future research adopts diverse methods for acquiring study data and integrates emerging EEG technologies as complementary data sources. Furthermore, considering the integration of EEG technology with human–computer interaction could meet the lifestyle needs of individuals with post-brain ischemia disabilities [208,209].

## 3. Recent Treatment Progress

### 3.1. Drug Therapy

Currently, chemical drugs, including benzodiazepines, non-benzodiazepines and the latest appetite receptor antagonists, are still the predominant therapeutic approach. Antidepressants, including mianserin, exhibited efficacy in ameliorating post-stroke insomnia [42,210]. Non-benzodiazepine medication, specifically zolpidem tartrate tablets, demonstrated effectiveness in enhancing neurological function, improving sleep quality and increasing 5-HT levels in individuals suffering from acute stroke-related sleep disorders [211,212]. Zolpidem exhibited a minimal effect on cognitive function and muscle tone, whilst potentially enhancing the prognosis of acute stroke by stimulating the secretion of brain-derived neurotrophic factor (BDNF) and safeguarding the integrity of the neurovascular unit [213,214]. Research indicated that zolpidem tartrate tablets do not negatively affect patients’ blood pressure [215,216,217]. OxA has been identified as a key driver of arousal and alertness [218]. OxA receptor antagonists, such as daridorexant, are a new class of hypnotics [219]. Daridorexant selectively impedes the interaction of the neuropeptide OxA with OxA receptors OX1R and OX2R, leading to the attenuation of wakefulness. According to findings from clinical trials, suvorexant may be a viable option for patients exhibiting unresponsiveness to treatments involving benzodiazepine and non-benzodiazepine hypnotics [220]. This novel prescription therapy enhances sleep parameters whilst exhibiting a reduced incidence of side effects compared with the traditional pharmacological methods [218].

However, in several studies, hypnotics have been found to increase the risk of stroke. Evidence suggests that levocetirizine (zolpidem) is associated with an increased risk of ischemic stroke, and that the risk increases with increasing doses. The annual use of over 4 g benzodiazepines or use of the drug for more than 95 d is known to increase the incidence of stroke [221]. OxA receptor antagonists are linked with increased costs, and they pose potential safety concerns attributed to restricted clinical usage. Therefore, a comprehensive evaluation of the risk-benefit equilibrium is crucial before contemplating the use of sedatives in the management of post-stroke sleep disorders.

### 3.2. Non-Pharmacological Treatment

Non-pharmacological treatments include psychotherapy, acupuncture and hyperbaric oxygen therapy. The emergence of sleep disorders post-stroke is inextricably linked to the patient’s psychological factors. The latest therapeutic approach, cognitive behavioral therapy (CBT) for insomnia (CBT-I), stands out as the most secure and efficacious treatment option, boasting a notable degree of practical applicability [222]. It systematically examines the cognitive processes of patients by assessing their behavior and emotional responses, pinpointing cognitive distortions and facilitating cognitive restructuring [223]. A study comparing CBT and conventional therapy in patients with post-stroke sleep disorders revealed that CBT demonstrated superior efficacy and yielded a notable improvement in sleep quality [224]. CBT may be employed for the long-term management of chronic insomnia in patients with stroke. It appears to be efficacious in mitigating post-stroke insomnia. However, significant long-term physical and psychological benefits were observed, and studies with larger samples are needed in the future to address such issues [11,224].

Multiple randomized controlled trials have shown that acupuncture serves as a useful tool for patients with post-stroke sleep disorders [225,226,227]. Recently, acupuncture therapy, as a traditional Chinese medicine therapy, has been widely used in the treatment of post-stroke sleep disorders for its convenient operation and low adverse effect characteristics [228,229,230,231]. The results of clinical studies have indicated that treatment with acupuncture is significantly effective in relieving patients’ symptoms and improving sleep quality and symptoms of neurological deficits [232]. In the efficacy evaluation, the total effective rate of the acupuncture treatment group reached 87.50% (84/96), which was significantly higher than that of the control group at 76.04% (73/96) [233]. A study via the Pittsburgh Sleep Quality Index Assessment found that patients with stroke and insomnia who received acupuncture tended to show more effective outcomes than those taking medication [234]. This finding suggested that acupuncture treatment has certain therapeutic advantages in the treatment of post-stroke sleep disorders, and all of these findings suggested that acupuncture is a promising approach for the treatment of post-stroke sleep disorders.

Hyperbaric oxygen could play a role in improving the symptoms of cerebral ischemia and regulating the function of the brain cortex, improving cellular hypoxia and protecting neurological function. Clinical studies have indicated that the efficacy of hyperbaric oxygen in the treatment of post-stroke insomnia is comparable to that of dexzopiclone, more long-lasting and effective in the recovery of neurological function [235].

### 3.3. Combination Therapy

Combination therapy for post-stroke sleep disorders has better-integrated outcomes. A study showed that 98 patients with sleep disorders post-stroke who were given sertraline in combination with hyperbaric oxygen therapy showed enhanced sleep quality scores, indicating that sertraline in combination with hyperbaric oxygen therapy has the potential to improve the quality of sleep in patients with sleep disorders post-stroke [236]. Similarly, the clinical effect of using haloperidol melittin tablets in combination with hyperbaric oxygen for the treatment of post-stroke sleep disorders was clinically found to be precise and improve sleep quality [237]. Studies using VD combined with dexzopiclone and escitalopram oxalate therapy effectively modulated inflammatory factors and neurocytokine levels in patients with post-stroke sleep disorders and improved the quality of their sleep [238]. Further studies found that the method increased the overall treatment efficiency (95.51%, 85/89), improved the level of cerebral perfusion indices and reduced PSQI and ESS scores and adverse effects [239]. Studies suggested that combining the Bailemian capsule (with 2,3,5,4′-tetrahydroxy stilbene-2-O-β-D-glucoside as the main active ingredient) with CBT-I could be an effective option for clinically treating post-stroke sleep disorders [223]. In addition, the application of acupuncture combined with Huanglian Wendan Decoction for the treatment of sleep disorders in stroke can effectively reduce the incidence of adverse reactions and promote the improvement of the treatment and prognosis of patients [240]. The total effective rate of therapy was as high as 94.00%, which was better than 76.00% in the control group, and the TST and serum BDNF, GDNF and NGF levels were higher after treatment than before treatment [241]. Combining different therapies for post-stroke sleep disorders offers a more effective and comprehensive treatment by addressing the issue from multiple perspectives simultaneously.

### 3.4. Emerging Therapies

We examined novel therapeutic approaches for managing post-stroke sleep disorders, alongside established strategies. In pharmacotherapy, suvorexant, lemborexant, and daridorexant stand out as the most advanced orexin receptor antagonists for treating insomnia. Numerous meta-analyses and clinical trial reviews support their effectiveness [242,243,244,245,246]. Considering its clinical efficacy, safety, and pharmacological properties, the study suggests that lemborexant 5 mg is the most suitable first-line treatment option [245]. Targeted appetitive hormone systems show promise in understanding and treating sleep disorders, but more studies are needed. Additionally, the efficacy difference between dual and selective antagonists also warrants investigation [247]. Cutting-edge bioelectrical signaling technologies, like brain–computer interfaces, offer potential for improving sleep quality [248]. These cutting-edge tools, employing advanced neuroscience and engineering techniques, offer entirely new possibilities for treating post-stroke sleep disorders.

Neuromodulation techniques, such as transcranial magnetic stimulation and transcranial electrical stimulation, offer viable therapeutic potentials [249]. Non-invasive brain stimulation therapies, including transcranial ultrasound-neuromuscular stimulation therapy and light therapy, show promise for insomnia patients during stroke recovery [250]. These modalities improve sleep quality and negative moods in patients. Studies show varied effects on EEG α rhythm modulation with different cranial direct current stimulation [251]. While meta-analyses support non-invasive brain stimulation therapy effectiveness in stroke treatment, future research should increase sample sizes and improve study quality [252,253]. Additionally, research indicates that targeting inflammatory vesicles through LED light stimulation could be a novel therapeutic approach to improve brain injury after ischemic stroke [254].

Virtual reality (VR) therapy, an advanced human–computer interface, has shown effectiveness in stroke rehabilitation by creating a tranquil, sleep-friendly environment through virtual settings [255]. This innovative approach offers a more immersive experience for patients and is anticipated to be crucial in treating sleep disorders post-stroke. Studies suggest that VR technology can enhance balance function, promote limb rehabilitation, and aid cognitive recovery, surpassing some traditional treatments [256,257]. However, its clinical application for improving insomnia in stroke patients is limited, and its overall value is not fully demonstrated.

## 4. Conclusions and Prospect

A large evidence base suggests that sleep disorders are strongly associated with incident strokes. This review focused on the main specific correlates of sleep disturbances caused after stroke, including functional alterations in brain regions, cerebrovascular hemodynamics, neurological deficits, interference with sleep ion regulation, neurotransmitter regulation and inflammatory response (Figure 1). Stroke disrupts sleep patterns, affecting sleep staging and rhythms, and the extent of this effect varies on the basis of the site of the lesion. Moreover, cerebral hemodynamics are disturbed post-stroke, leading to sleep issues. Stroke may disrupt sleep by directly or indirectly influencing sleep-related ion concentrations (K^+^, Ca^2+^ and Mg^2+^), ion channels (TREK-1 and GIRK channels), and the expression levels of kinases (ERK1/2, CaMKII/β and SIKs). Stroke notably disrupts the secretion of sleep-related neurotransmitters by reducing levels of 5-HT and increasing the levels of OxA. Additionally, it explicitly increases the levels of inflammatory factors, including IL-6, CRP, IL-1β, TNF-α, Hcrt and others. This review also focused on the recent treatment progress. Sedative antidepressants are recommended for the treatment of post-stroke sleep disorders given the increasing supporting evidence. However, behavioral neuropsychological therapies and other supportive therapies (e.g., acupuncture) are prospective, but they still need to be backed up by more evidentiary support.

To date, multiple studies have explored different types of stroke sleep disorders and suggested potential treatment strategies and pitfalls of these treatments to provide an overview of the different types of stroke sleep disorders. However, lingering questions remain. Studies of sleep disorders in animal models of stroke could contribute to further insights into how the brain controls sleep and wakefulness, which could potentially reveal more pathophysiological mechanisms behind sleep disorders and the complex relationship between stroke and sleep processes. A strategy for treating post-stroke-related sleep disorders is expected to be developed with the integrated application of animal models and neuronal modelling approaches on the basis of computational models of sleep disorders. Studies have confirmed that stroke-related sleep impairment negatively regulates angiogenesis, axonal sprouting and synaptogenesis, exacerbating brain damage and ultimately impeding neurological recovery. However, the existing clinical evidence remains relatively restricted regarding the causal relationship between sleep disorders and stroke recovery.

In the future, the authors confidently believe that sleep cells in NVUs with multiple associations play important roles in cerebral ischemia and sleep. Hence, thoroughly exploring the mechanism of ion channel action in NVU post-stroke is highly important. A strategy for treating post-stroke-related sleep disorders is expected to be developed with the help of a combination of animal models and neuronal modelling approaches on the basis of computational models of sleep disorders. A deepened understanding of the interrelationship between circadian rhythms and stroke changes is critical for the early identification and management of stroke-specific changes.

Sleep disorders in patients stroke could receive more attention, and more personalized approaches could be developed. In addition, more studies are needed to validate the long-term efficacy and safety of these cutting-edge approaches. Continuing innovations in this field offer compelling prospects for future developments in treating post-stroke sleep disorders. Focusing on sleep patterns and their interference, including genetic and neurochemical variations, could contribute to an enhanced understanding of the core causes of a patient’s sleep issues. Many critical questions concerning the relationship between sleep disorders and stroke remain largely unknown, and more energy needs to be devoted to them.

## Figures and Tables

**Figure 1 brainsci-14-00307-f001:**
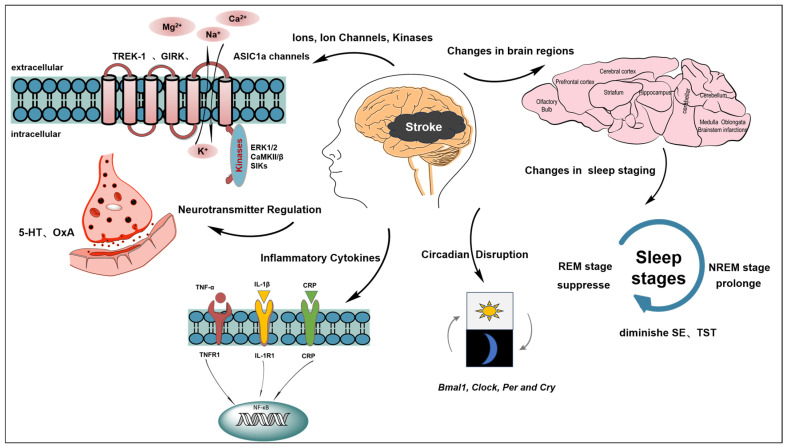
Mechanisms of sleep disorders caused by stroke. Stroke alters sleep patterns by affecting staging, rhythms, and is influenced by the lesion site. Cerebral hemodynamics post-stroke also contribute to sleep disturbances. Direct or indirect influences include disruptions to sleep-related ion concentrations (K^+^, Ca^2+^, Mg^2+^), ion channels (TREK-1, GIRK), and the expression of kinases (ERK1/2, CaMKII/β, SIKs).

**Figure 2 brainsci-14-00307-f002:**
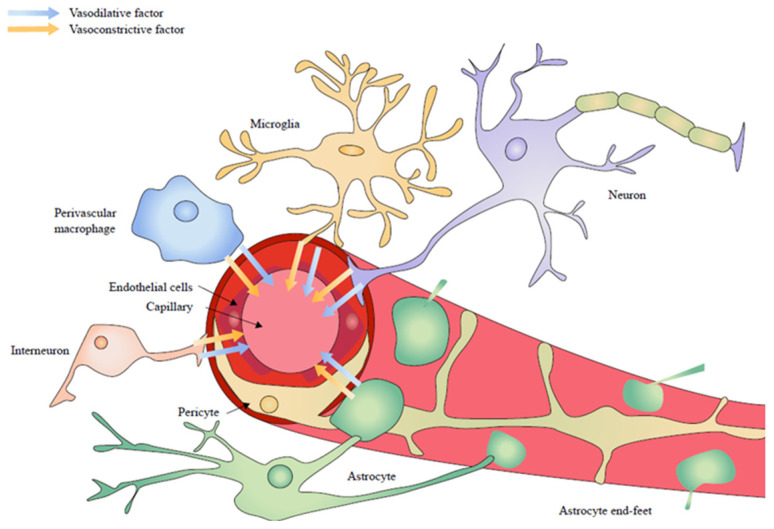
Neurovascular unit, encompassing endothelial cells, pericytes, astrocytes, neurons, endogenous neurons, perivascular macrophages and peripheral microglia.

**Table 1 brainsci-14-00307-t001:** Site of lesion, sleep staging and rhythmic changes.

Type of Lesion	Site of Lesion	Changes in Sleep Structure and Rhythm	References
stroke	anterior cerebral artery area infarctions	diminished β waves and increased δ waves increase slowing in θ ranges rhythms decreased overall amplitude	[14]
stroke	preoptic	reduced NREM and REM	[47]
stroke	thalamic	decreased sleep spindles increased N1 and decreased N2	[49]
stroke	supratentorial stroke	reduced NREM TST, low SE	[50]
acute stroke	lenticulostriate arteries MCA cortical branches	reduced REM	[20]
acute hemispheric stroke	hemispheric	TST, low SE, reduced N2 and decreased N3 and N4 NREM sleep	[51]
raphe nucleus stroke	raphe nucleus	reduced NREM	[6]
cerebral hemorrhagic infarction	frontal lobe	increased δ waves	[52]
brain stem strokes	brainstem	highest REM and REM latency	[53]
brain stem stroke	thalamus mesencephalic pontine tegmental reticular formation	diminished REM sleep increased NREM sleep	[54]
ischemic stroke	cortex and striatum	inhibited REM sleep	[55]
cerebellar stroke	brain stem and hemisphere	reduced NREM prolonged REM latency	[56]
paramedian thalamic stroke	paramedian thalamic	increased N2 and N3 decreased N4	[57]
bilateral thalamic stroke	bilateral thalamic	NREM sleep instability reduced arousals	[58]
unilateral diencephalic stroke	thalamus	excessive sleep decreased N2 and N3 sleep	[59]
lateral medullary infarction	lateral medullary	complete sleep suppression	[60]
middle-aged C57BL/6J mice, MCAO	frontoparietal cortex and lateral caudoputamen	reduced NREM and REM increased latency to sleep reduced NREM delta power	[61]

TST = total sleep time; SE = sleep efficiency; MCAO = middle cerebral artery occlusion.

## Data Availability

Not applicable.

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
