# Peer review of "Deciphering Post-Stroke Sleep Disorders: Unveiling Neurological Mechanisms in the Realm of Brain Science"

_brainsci, 2024, doi:10.3390/brainsci14040307_

Round 1

Reviewer 1 Report

Comments and Suggestions for Authors

The article aims to conduct a systematic review of scientific publications to elucidate the mechanisms of stroke-induced sleep disturbances. Thus, the purpose of the article is clearly stated.

The article covers an important gap in scientific knowledge. It does so by elucidating the reciprocal relationships and novel biological mechanisms that link post-stroke sleep abnormalities with cerebral ischemia.

Publication methodology is not described in the article. The authors should clearly describe the algorithm of selection and selection of publications. The PRISMA method should be used. It is necessary to specify which databases and keywords were analyzed. Nevertheless, the authors have done a lot of work with literature sources. All used sources are relevant.

The authors use interesting illustrations in the article. The authors should cite their source. If the authors created the illustrations themselves, it is worth mentioning what software or resources were used for this.

Reviewer 2 Report

Comments and Suggestions for Authors

The present manuscript it of great importance for doctors involved both in acute and chronic phase of stroke care. The topic is up-to-date and the authors present the data in a comprehensive and detailed manner.
The topic is well formed giving the exact accent of the manuscript. The abstract is fit and following all requirements. The introduction is stressing out on the problem and giving additional information about the possible mechanisms for sleep disorders. The body of the manuscript gives detailed information about the various mechanisma and topics of the lesion related with sleep disturbances. A great aspect of the manuscroipt is the therapeutical approach discussion, but not only focusin on ethilogy and pathogenesis. The conclusions follow the main course of the manuscript and are consistable with everything above mentioned. In addition the author's team present a good visualization (figures) in the manuscript increasing the quality of the work.
In general I do have no further recommendation for improving the quality of the manuscript. The author's team did a great job, which is visible from the large amount of data they proceeded, and the final result they are presenting.

Comments on the Quality of English Language

The quality of the language is good, but yet another run though the manuscript is advisable in order to figure out some technical issues.

Reviewer 3 Report

Comments and Suggestions for Authors

Clarification of Methodological Approach: The paper could benefit from a clearer explanation of the methodologies used in synthesizing data from the reviewed literature. Specifically, outlining the criteria for selecting studies, the process of data extraction, and how findings were aggregated or compared would enhance the reader's understanding and confidence in the review's conclusions.

Discussion of Limitations: While the paper provides a comprehensive overview of post-stroke sleep disorders, a more detailed discussion of the limitations of the current body of research and how these limitations impact the conclusions that can be drawn would strengthen the paper. This includes limitations related to study design, variability in measurement tools for sleep disorders, and potential publication bias.

Expansion on Therapeutic Strategies: The section on potential therapeutic strategies for addressing post-stroke sleep disorders is informative but could be expanded to include more detail on emerging therapies, their mechanisms of action, and the evidence supporting their efficacy. Additionally, discussing the implications of these therapies for clinical practice and future research directions would provide valuable context for readers.

Round 2

Reviewer 1 Report

Comments and Suggestions for Authors

Dear authors,

Thank you very much. All the changes are approved.